# Model-based reconstruction of whole organ growth dynamics reveals invariant patterns in leaf morphogenesis

Mohamed Oughou[1], Eric Biot[1] , Nicolas Arnaud[1] , Aude Maugarny-Calès[1,2] , Patrick Laufs[1] , Philippe Andrey[1] , and Jasmine Burguet[1]

[1]Université Paris-Saclay, INRAE, AgroParisTech, Institut Jean-Pierre Bourgin (IJPB), 78000 Versailles, France;
[2]Université Paris-Saclay, 91405 Orsay, France

## Original Research Article

Arabidopsis thaliana; growth trajectory; organ initiation modeling; plant development; shape quantification; spatio-temporal analysis.

**Authors for correspondence:**
Jasmine Burguet,
E-mail: Jasmine.Burguet@inrae.fr

## Abstract

Plant organ morphogenesis spans several orders of magnitude in time and space. Because of limitations in live-imaging, analysing whole organ growth from initiation to mature stages typically rely on static data sampled from different timepoints and individuals. We introduce a new model-based strategy for dating organs and for reconstructing morphogenetic trajectories over unlimited time windows based on static data. Using this approach, we show that *Arabidopsis thaliana* leaves are initiated at regular 1-day intervals. Despite contrasted adult morphologies, leaves of different ranks exhibited shared growth dynamics, with linear gradations of growth parameters according to leaf rank. At the sub-organ scale, successive serrations from same or different leaves also followed shared growth dynamics, suggesting that global and local leaf growth patterns are decoupled. Analysing mutants leaves with altered morphology highlighted the decorrelation between adult shapes and morphogenetic trajectories, thus stressing the benefits of our approach in identifying determinants and critical timepoints during organ morphogenesis.

## 1. Introduction

Morphogenesis involves intricate mechanisms that operate in time and space in a coordinated manner to produce stereotyped organs with common morphological characteristics. Significant morphological changes may occur early during organ development. This is the case for plant leaves, whose shape can undergo major morphological changes when the organ is only few hundreds of micrometers long, including the initiation of marginal outgrowths, like teeth, whose pattern and size form major characteristics of mature leaf shape (Biot et al., 2016). Time-lapse microscopy is the tool of choice for the spatiotemporal monitoring of organ growth at early developmental phase. State-of-the-art acquisition techniques allow following sample development only up to few days for animals (Berger et al., 2021; Park et al., 2015; Zattara et al., 2016), organoids (Hof et al., 2021), plant leaves (Kierzkowski et al., 2019; Serra and Perrot-Rechenmann, 2020) or floral organs (Fox et al., 2018; Rambaud-Lavigne and Hay, 2020; Ripoll et al., 2019). However, long-term observations are hampered by phototoxicity, photobleaching, tissue heating and manipulations that may significantly impact morphogenesis and increase lethality (Bell, 2017). To analyse and quantify phenotypes, phenomics propose a panel of techniques including imaging to follow developing systems (Tardieu et al., 2017). However, they mainly address whole organism scale, and cannot capture morphological changes of very early developmental phases. Organ monitoring over long periods is indeed made difficult by huge changes in size, which require adjusting the observation protocols from microscopic to macroscopic scales. A convenient alternative is the use of static data collected on different individuals at different developmental timepoints. More samples can be observed than those typically provided by live-imaging approaches, thus allowing better assessment of variability. Another advantage is that the biological system evolves in normal conditions till sampling, observation and measurement. In return, methods for reconstructing continuous developmental trajectories from static observations are required.

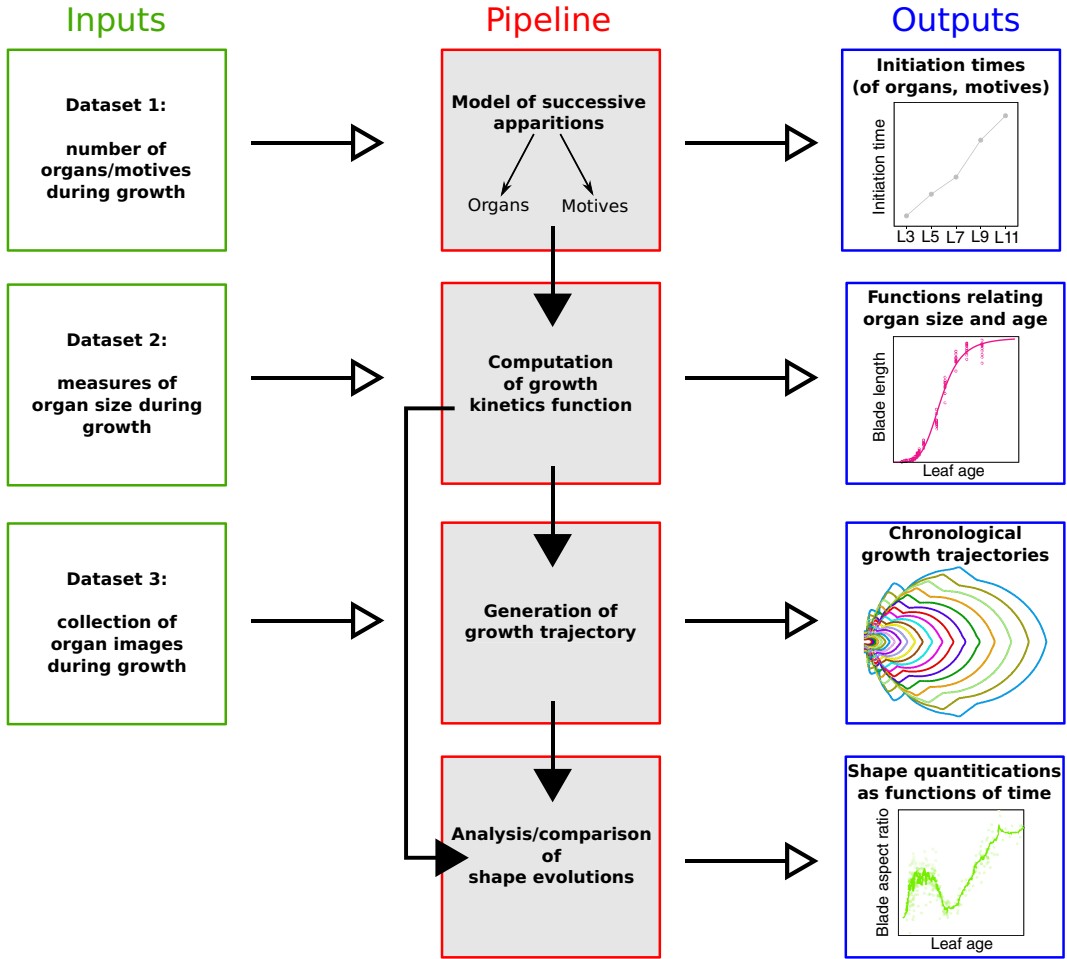

**Fig. 1.** Overview of the proposed methodology for the spatiotemporal reconstruction and quantification of organ morphogenesis based of collections of static data. *Left*: independent sets of static data feeding different stages of the pipeline. *Middle*: steps for dating static individual organs and quantifying the continuous shape evolution during growth. *Right*: outputs obtained at the different levels, that can be used independently to analyse morphogenesis.

Reconstructing developmental trajectories implies accurately dating the specimens experimentally observed. Yet, it is generally difficult to assign an age to an organ, because the time at which it was initiated is unknown. In the case of leaves, a popular solution is the plastochron index (PI) (Erickson and Michelini, 1957; Meicenheimer, 2014) or the related leaf PI (LPI), which define a standardised morphological age instead of a chronological one. However, PI/LPI methods assume exponential growth, equal growth rates between organs, and constant plastochron (delay separating the apparition of two successive organs). These requirements must be a priori tested, and are not systematically fulfilled. As an alternative, organ size may be used as a convenient proxy for developmental time. This approximation was successfully employed to analyse leaf development (Gonçalves et al., 2017; Maugarny-Calès et al., 2019) using MorphoLeaf software developed in our lab (Biot et al., 2016). However, this approach is limited because growth is not always linearly related to time and because it does not allow comparing growth dynamics of organs developing at distinct growth rates. In addition, it is preferable to measure development as a function of chronological time instead of morphological one. Neither PI/LPI nor size as a temporal proxy provide explicit temporal information in standard time units.

Because the analysis of complete organ morphogenesis over a long period is difficult to achieve with live imaging, we propose a new approach to reconstruct and quantify morphogenesis as a function of time from static data collected on different individuals (see Figure 1 and Supplementary Figure S1). As previously mentioned, this implies being able to assign an age to any sampled organ. For this, we first introduce a mathematical model of the sequential apparition of organs that we apply to estimate initiation times, thus allowing the temporal registration of growth dynamics between organs. Second, we use calibration curves relating organ age and size to estimate the ages of samples collected at arbitrary timepoints. MorphoLeaf software was modified to integrate this temporal calibration in the reconstruction of developmental trajectories, thus providing shape analyses as continuous functions of chronological time. We used this methodology to analyse the morphogenesis of a series of rosette leaves in *Arabidopsis thaliana*, whose size and shape significantly evolve according to apparition rank (Hunter et al., 2006; Tsukaya et al., 2000). Our results reveal invariant developmental patterns at both global (blade) and local (tooth) scales, that were hidden by contrasted sizes and shapes. Our study also shows uncoupling between tissue growth at global and local scales. Finally, we illustrate how our method allows dissecting

the spatiotemporal effects of mutations on morphogenesis, thus contributing to deciphering the molecular basis of morphogenetic processes.

## 2. Materials and methods

### 2.1. Plant material

All considered wild-type and mutant *Arabidopsis thaliana* plants were from the Columbia-0 (Col-0) ecotype and grew in controlled-environment rooms, in short-day conditions [1 hr dawn (19°C, 80 μmol m$^{-2}$ s$^{-1}$ light), 6 hr day (21°C, 120 μmol m$^{-2}$ s$^{-1}$ light), 1 hr dusk (20°C, 80 μmol m$^{-2}$ s$^{-1}$ light) and 16 hr dark (18°C, no light)]. Rosette leaves of odd ranks from 3 to 11 (denoted L3, L5, L7, L9 and L11 in the sequel) were analysed. In addition, we also analysed L11 in the *kluh-4* (Anastasiou et al., 2007) and *clf-81 sep3-2* (Lopez-Vernaza et al., 2012) Col-0 mutants (here denoted *kluh* and *clfsep*, respectively), which both present alterations in leaf morphology. In the following, sowing was considered as plant initiation time. Samples were randomly collected at different timepoints.

Three different datasets were generated to feed our quantitative analysis pipeline (see Figure 1). The first one was used to estimate leaf initiation time for each leaf rank (Dataset 1). Plants were dissected and observed using a binocular microscope (Nikon SMZ645) to determine the number of leaves, including the smallest visible primordium. This ensured that leaves were counted as soon as they appeared as primordia at the periphery of the shoot apical meristem. For each genotype, the output was a collection of pairs $(a_p, n)_i$, with $a_p$ the plant age when dissected and $n$ the number of observed leaves for any individual plant $i$. Leaf number was determined in 295, 33 and 38 plants for Col-0, *clfsep* and *kluh*, respectively.

The second dataset was used to estimate the temporal dynamics of leaf size (Dataset 2). Each leaf was dissected using a binocular microscope (Nikon SMZ645), its blade digitized using an Axio Zoom V16 microscope (Carl Zeiss Microscopy) with magnification depending on leaf size (from 125x to 8x) or, in case of a mature leaf, using a Perfection V800 Photo scanner (Epson), and then its length manually measured using Fiji software (Schindelin et al., 2012). Blade length was retained to quantify organ size because it is easily accessible and does not require the segmentation of leaf images as would be the case with leaf area. We thus obtained, for each leaf rank $r$ in a given genotype, a collection of pairs $(a_p, l)_i$, where $a_p$ is plant age and $l$ is blade length of individual plant $i$. Blade lengths were measured for Col-0 in 201, 227, 200, 167 and 117 leaves of ranks 3, 5, 7, 9 and 11, respectively, and in 390 and 229 L11 for *clfsep* and *kluh*, respectively.

The third dataset was used for quantitative image analysis and reconstruction of growth trajectory in MorphoLeaf software (Dataset 3). For this, wild-type data previously generated were reused and enriched following the same acquisition protocol as previously (Biot et al., 2016). Using MorphoLeaf, following automatic leaf contour segmentation, the proximal limit of the blade contours were manually delineated and morphological landmarks were automatically computed (Biot et al., 2016). Manual corrections were applied if necessary. This yielded five sets of 2D images for Col-0 L3, L5, L7, L9 and L11, with 160, 196, 168, 162 and 312 images, respectively, with organ length ranging from several tens of microns up to few centimetres for mature leaves. Images of L11 were similarly generated for the mutants, providing 214 and 390 images for *kluh* and *clfsep*, respectively.

### 2.2. Growth curves

A method was setup to estimate the function relating organ size and age. Let us consider a given leaf in the rosette (e.g., L11 in Col-0), for which initiation time $T$ was previously determined (see Section 3), and observed pairs $(a_p, l)_i$ associating plant age and blade lengths (Dataset 2). Initiation time $T$ was used to translate these data in time, to obtain pairs $(a, l)_i$, where $a = a_p - T$ is the age of the leaf since its appearance in plant $i$. For any organ considered in this study, length evolution with time exhibited a sigmoid shape. The Hill model was the most efficient in the family of sigmoid functions to fit size measures (Supplementary Figure S5). The function $\mathcal{L}$ relating length $l$ and age $a$ of the leaf from initiation was thus given by

$$\mathcal{L}(a) = L_\infty \frac{a^n}{t_{50}{}^n + a^n}, \qquad (1)$$

with $L_\infty$ the length of the mature leaf, $t_{50}$ the time at which the length is half of $L_\infty$, and $n$ the Hill parameter. These parameters were estimated using non-linear least-square regression. The contribution of each observation to the cost function was weighted by the inverse variance computed at the corresponding timepoint.

Several growth features were computed from the fitted Hill function. The time where maximal growth was reached (inflection point) is given by

$$t_{\max} = t_{50} \left( \frac{n-1}{n+1} \right)^{\frac{1}{n}}.$$

The maximal growth rate is given by

$$\mathcal{L}'(t_{\max}) = \frac{L_\infty}{4nt_{50}} (n+1)^2 \left( \frac{n-1}{n+1} \right)^{\frac{n-1}{n}},$$

where $\mathcal{L}'$ is the derivative of $\mathcal{L}$, and the relative growth rate at the time $t_{\max}$ of maximal growth is obtained as

$$\frac{\mathcal{L}'(t_{\max})}{\mathcal{L}(t_{\max})} = \frac{n+1}{2t_{50}} \left( \frac{n+1}{n-1} \right)^{\frac{1}{n}}.$$

The inverted fitted Hill function was subsequently used to estimate the age of any leaf from its experimentally measured length $l$:

$$\widehat{a}(l) = t_{50} \left( \frac{l}{L_\infty - l} \right)^{\frac{1}{n}}. \qquad (2)$$

### 2.3. Morphogenesis quantification and reconstruction

MorphoLeaf software was used to quantify leaf blade shape evolution during growth, from collections of 2D organ images (Dataset 3). Using segmented leaf contours and morphological landmarks, it provides measures of both global shapes (e.g., leaf blade area or elongation) and serrations at the margin (e.g., tooth area or aspect-ratio). The application also generates a mean growth trajectory, that is, a sequence of organ contours computed by time-weighted averaging of individual contours (Biot et al., 2016).

Blade length was used as a proxy of leaf age in the initial MorphoLeaf release (Biot et al., 2016). Here, we modified the application to integrate real organ age. Using the estimated parameters of the growth function ($L_\infty$, $n$ and $t_{50}$), MorphoLeaf software computes the age of each individual organ from its length [equation (2)]. We also improved the robustness of the computed growth trajectories with regards to sparsity, non-uniformity and local asymmetry in the temporal sampling of the data (see Supplementary Figure S2 for details).

## 2.4. Numerical and statistical methods

Systems of differential equations were simulated within the COPASI software (Hoops et al., 2006) using the RADAU5 integration method with default initial step-size and tolerance parameters. Model parameters were estimated in COPASI using the Evolutionary Strategy with Stochastic Ranking method (Runarsson and Yao, 2000) with default parameters. The cost function minimised in the estimation procedure was the sum of squared differences between predicted $(N_0, \ldots, N_K)$ and experimentally obtained $(M_0, \ldots, M_K)$ numbers of individuals with $0, \ldots, K$ organs at any time in the population.

The experimental dynamics of the numbers $M_0(t), \ldots, M_K(t)$ of individuals having $0, \ldots, K$ organs were not directly observed. Indeed, our experimental data (e.g., Dataset 1 for the number of leaves per plant) were sets of couples $t_i, k_i$, where $t_i$ was the age of the $i$th individual and $k_i$ its number of organs. We therefore devised a procedure to infer the empirical population dynamics from our experimental recordings.

We assumed ergodicity and considered our observations corresponded to the temporal evolution of a population of $N$ individuals. The initial data were resampled to ensure the same number of observations was present in all categories. $N$ was set to the size of the most represented category in the initial data.

For the category of individuals with no organ ($k = 0$), we reasoned that since all individuals eventually acquire organs, $M_0(t) = 0$ for $t$ sufficiently large. Moving backwards in time from infinity, $M_0$ was increased by +1 each time an individual with $k_i = 0$ was observed. This led to

$$M_0(t) = \#\{k_i = 0 \wedge t_i > t\}$$
$$= N - F_0(t),$$

where $F_k(t)$ is the cumulative number of observed individuals in category $k$ up to time $t$:

$$F_k(t) = \#\{k_i = k \wedge t_i \leq t\}.$$

For arbitrary $0 < k < K$, we followed the same line of reasoning, taking into account the individuals that may still be in the previous category at any time. Hence, starting from $M_k(t) = 0$ for $t$ at infinity and moving backwards in time, $M_k$ was increased by +1 each time an individual was observed in category $k$ and decreased by -1 each time an individual was observed in category $k-1$. This yielded:

$$M_k(t) = \#\{k_i = k \wedge t_i > t\} - \#\{k_i = k - 1 \wedge t_i > t\}$$
$$= N - F_k(t) - [N - F_{k-1}(t)]$$
$$= F_{k-1}(t) - F_k(t).$$

For $k = K$, we have $M_K(t) = N$ for $t$ sufficiently large. At any time, the individuals still present in the previous category were removed from $M_K$:

$$M_K(t) = N - \#\{k_i = K - 1 \wedge t_i > t\}$$
$$= N - (N - F_{K-1}(t))$$
$$= F_{K-1}(t).$$

## 3. Results

### 3.1. A mathematical model of organ apparition dynamics

We introduce a mathematical model of the population dynamics of organ apparition. Let us consider a population of $N$ individuals that acquires organs sequentially with time. The master equation in the model describes the instantaneous variation of the number $N_k(t)$ of individuals having $k$ organs at time $t$:

$$\frac{dN_k(t)}{dt} = \alpha_{k-1}(t)N_{k-1}(t) - \alpha_k(t)N_k(t),$$

where $\alpha_k(t)$ is the quantity of individuals with $k$ organs that acquire a new organ between $t$ and $t + dt$. This equation captures the fact that individuals pass successively through categories with increasing numbers of organs, each category feeding the next one.

Two additional equations complete the model for the extreme categories. For individuals with no organ, we have

$$\frac{dN_0(t)}{dt} = -\alpha_0(t)N_0(t),$$

and for individuals having reached the maximal number $K$ of organs, we have

$$\frac{dN_K(t)}{dt} = \alpha_{K-1}(t)N_{K-1}.$$

Since all individuals have initially no organ, the initial conditions are

$$\begin{cases} k = 0 : & N_0(0) = N, \\ k > 0 : & N_k(0) = 0. \end{cases}$$

To complete the definition of the model, an explicit form must be chosen for the transition functions $\alpha_k$, ensuring that the transitions are triggered sequentially. To this end, we choose sigmoid functions

$$\alpha_k(t) = \frac{s_k}{1 + \exp[-\beta_k(t - t_k)]},$$

where $\beta_k$, $s_k$ and $t_k$ are free parameters of the model. When fitting the model to experimental data, there were thus $3K$ parameters to be estimated. Parameter $t_k$ controls the time at which the transition between $k$ and $k + 1$ organs occurs (inflection point), $\beta_k$ controls the slope of the transition function, while $s_k$ is a scaling parameter.

To illustrate the behavior of the model, we considered an arbitrary situation with $K = 5$, setting the inflection points of the transition functions at increasingly spaced intervals. The obtained dynamics showed the capacity of the model to account for the sequential apparition of organs (Supplementary Figure S3).

### 3.2. Temporal dynamics of leaf initiation

We used our mathematical model to estimate the initiation times of leaves of different ranks (organ apparition orders in the plant) in Col-0. Using experimental recordings of the number of leaves as a function of plant age (Figure 2a; Dataset 1, see Section 2), we computed the experimental dynamics of the proportions of plants with varying numbers of leaves in a typical population (Figure 2b, *dots*). The same procedure was applied to mutant data (Supplementary Figure S4). Leaves of odd ranks only were considered in the three genotypes. In the wild type, we modelled the dynamics up to L11. In the mutants, we considered the dynamics from L9 to L11 (*kluh*) or from L7 to L11 (*clfsep*) to comply with the experimental samplings (Supplementary Figure S4 A,B).

The fit between model predictions and experimental dynamics was close to perfect, for both wild-type (Figure 2b) and mutant plants (Supplementary Figure S4). For each rank, leaf initiation time was estimated as the first time where the corresponding proportion had reached 50% of the population. The retained criterion for estimating initiation times from the model was not determinant as alternative criteria produced almost identical values

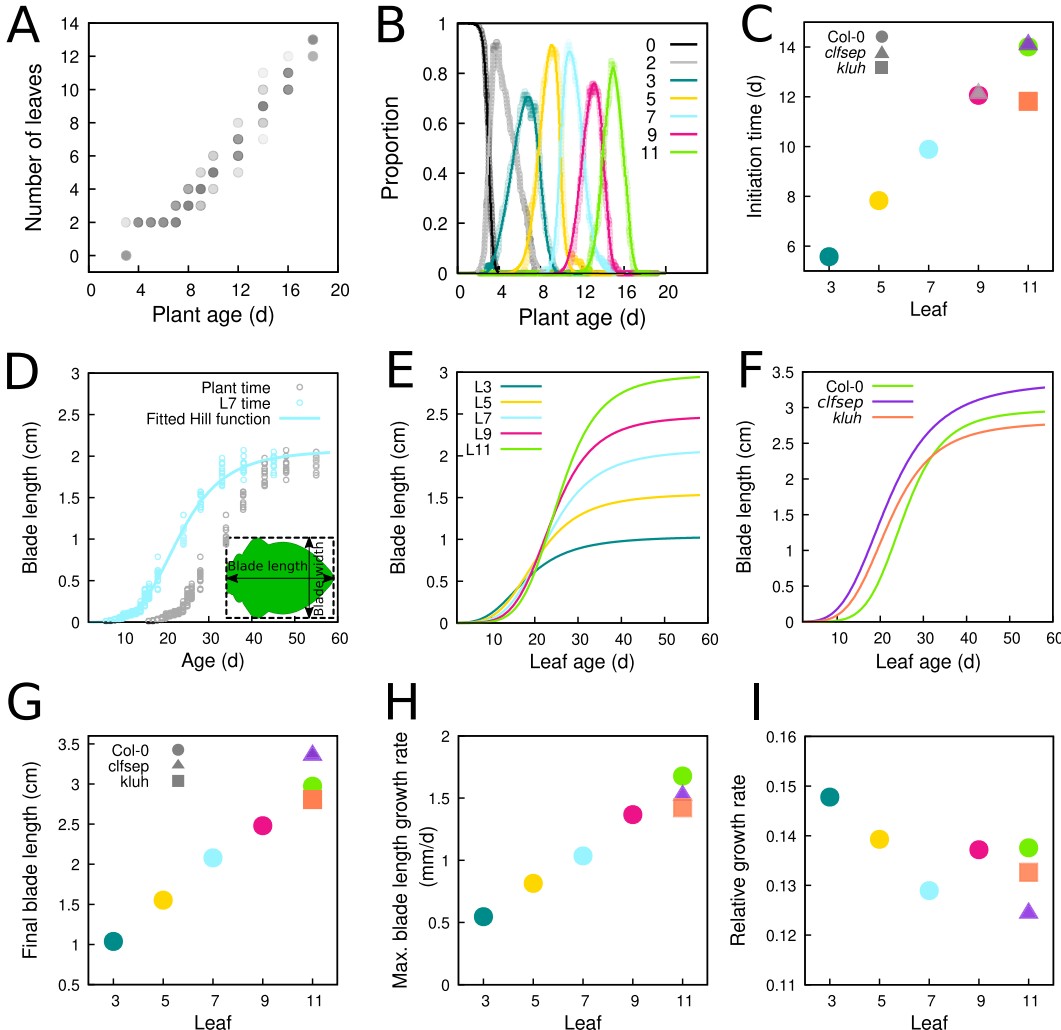

**Fig. 2.** Reconstructing leaf initiation and growth dynamics from static data. (a) Evolution of the number of rosette leaves during plant growth in Col-0 (*n*=345; opacity proportional to data density). (b) Temporal dynamics of empirically-derived and model-predicted proportions of individuals with different numbers of leaves in wild-type. *Dots*: empirical proportions computed from data in panel (a); *Curves*: model-predicted proportions after model fitting. (c) Estimated leaf initiation times for Col-0 in hours (*Circles*: odd ranks from 3 to 11, same colour coding as in panel (b), *clfsep* (*Triangles*: L9 and L11) and *kluh* (*Square*: L11). (d) Temporal dynamics of blade length for Col-0 L7. *Dots*: experimentally measured blade length expressed as a function of plant age (*Grey*) or leaf age (*Light blue*). *Continuous curve*: fitted Hill function. (e) Estimated growth dynamics of Col-0 leaves (fitted Hill functions). (f) Estimated growth dynamics for leaf 11 in Col-0, *clfsep* and *kluh*. (g–i) Evolution of growth parameters for the wild-type and the two mutants: final blade length ($L_\infty$) (g), maximal blade length growth rate at inflection point $t_{max}$ (h) and relative growth rate at inflection point (i). Ages are indicated in days (d).

(Supplementary Table S1). Plotting initiation times as a function of leaf rank showed that wild-type organs emerged at regular intervals of about 2 days between successive odd-ranked leaves (Figure 2c), suggesting a new organ emerged every day. There was, however, a globally decreasing trend in the interval between successive leaves, suggesting an acceleration in the apparition of organs with time (Supplementary Table S2). In *kluh*, L11 emerged about 2 days earlier than in the wild type, in agreement with previous reports of a shortened plastochron in this mutant (Wang et al., 2008). In *clfsep* mutant, leaves 9 and 11 appeared approximately at the same time as in Col-0. Overall, these results show how our dynamical model can be used to obtain fine parameter estimates of organ initiation dynamics from static observations.

### 3.3. Temporal dynamics of global leaf growth

To translate blade length measures (Dataset 2) from plant age to leaf age, we first applied a temporal shift using estimated organ

initiation time (Figure 2d). The functions relating leaf size and age were then evaluated for wild-type and mutant leaves (Figure 2d–f and Supplementary Figure S6).

The resulting estimated functions revealed a strongly patterned, regular arrangement of leaf growth kinetics according to organ rank in Col-0 (Figure 2e). There was a switch during growth in the ordering of leaf sizes of different ranks, while the final blade length increased with leaf rank, the reversed ordering was observed in the initial phase of growth. In addition, the graphs suggested that both the maximal growth rate and the time where it was reached also increased with leaf rank. Plotting parameters from the fitted Hill functions as a function of leaf rank quantitatively confirmed these trends in final leaf blade length (Figure 2g), maximal blade length growth rate (Figure 2h) and time at maximal blade length growth rate $t_{max}$ (Supplementary Figure S7). In contrast, there was no coherent gradient between leaf rank and relative growth rate at time of maximal growth (Figure 2i), showing differences in maximal growth were mainly due to size differences. Altogether, these results

suggest that successive leaves in the wild type follow a common developmental program that is differently parameterised according to organ rank, growth of later leaves appearing homothetically related to that of earlier ones, with a dilation in both time and space.

L11 growth dynamics differed between the three genotypes (Figure 2f). As observed between successive leaves in Col-0, there was a switch in size depending on early and late phases between the wild-type and *kluh* L11, the mutant leaf being initially longer and finally smaller. Therefore, growth dynamics of *kluh* L11 resembles the one of wild-type leaves of rank < 11, which is in agreement with an earlier initiation time of L11 in *kluh* compared to the wild type. On the other hand, *clfsep* L11 was always longer than the one in Col-0, suggesting the switch in size is not systematic. The uncoupling between growth patterns at different timepoints was further highlighted by comparing the parameters of fitted Hill functions. For example, the two mutations had opposite effects on final size, with *clfsep* L11 being about 13% larger and *kluh* L11 about 6% smaller than Col-0 L11 (Figure 2g), but they similarly induced earlier times at maximal growth (Supplementary Figure S7) and reduced maximal growth (Figure 2h).

Overall, these results illustrate the importance of considering whole temporal dynamics when analysing organ growth, as relative patterns in initial growth, maximal growth and final size may be uncoupled.

### 3.4. Leaf morphogenesis trajectories

We next examined the morphological changes that accompanied growth dynamics. Following the delineation of leaf contours and extraction of geometrical landmarks on leaves collected at different timepoints (Dataset 3, see Section 2), a continuous morphological trajectory was reconstructed for each leaf rank using a new version of the MorphoLeaf software (Figure 3a). The age since initiation of all collected leaves was computed using automatically measured blade lengths and growth kinetics functions.

In a previous work, we compared developmental trajectories between leaves of different ranks by analysing stages of the same length (Biot et al., 2016). This was informative but growth kinetics functions (Figure 2e) showed this induces comparing leaves at different ages. Here, taking into account the non-linear relationship between length and time referentials unmasked unsuspected differences between leaves of the same age. Comparing Col-0 leaves at days 10 and 26 after their initiation corroborated at the morphological level the temporal differences in relative sizes. For instance, the two final teeth on both sides of L3 were well-formed at day 10 after leaf initiation while at the same age, the first tooth of L11 (a leaf that ultimately carries at least four teeth on each side, two more than L3) was barely present (Figure 3b). Altogether, the elaboration of organ shape was delayed in a rank-dependent manner, as for global growth. Overall, these results show how age-based developmental trajectories provide new insights into the developmental origin of leaf heteroblasty, thanks to an increased temporal accuracy.

To quantitatively analyse shape dynamics along the reconstructed trajectories, we measured blade morphology parameters all along leaf growth. After having verified that growth trajectories accurately estimate leaf morphological traits (Supplementary Figure S11), we analysed global elongation and contour complexity using the blade aspect-ratio (length/width) and dissection index (blade convex hull area/blade area), respectively. For all considered ranks, the blade initially elongated and then became more round,

before elongating again after having reached about 2 mm in length (Figure 3c, *Left*, and Supplementary Figure S8). A transient increase in shape complexity was also observed for all leaves (Figure 3d, *Left*). Expressing shape measures as a function of leaf age (Figure 3c,d, *Middle*) quantitatively confirmed the temporal shift in developmental dynamics between successive leaves. These results further support the possibility that leaves develop according to a graded declination of a common morphogenetic module.

Both mutants exhibited more elongated L11 blades compared to Col-0 (Figure 3a and c, *Right*). This could be explained by the absence of the transient phase of rounding observed in Col-0. In addition, *kluh* displayed smoother final contours while teeth appeared to be more pronounced in *clfsep* (Figure 3b and d, *Right*). However, both mutants exhibited more developed teeth at day 10 (Figure 3b) and more precocious shape dynamics (Figure 3c,d, *Right*) compared with Col-0. These results are consistent with the observed earlier growth in these mutants and further highlight the uncoupling between different developmental features.

### 3.5. Dating teeth

To finely dissect the changes occurring during leaf morphogenesis, we next focused on individual teeth. To determine teeth initiation times, we applied the same strategy as the one used above for leaves. The number of teeth per half-leaf as a function of leaf age (Figure 4a, and Supplementary Figures S9 and S10) was automatically determined with MorphoLeaf from segmented leaf contours of Dataset 3. We then fitted our mathematical model of initiation dynamics to the empirically derived proportions of leaves with different numbers of teeth.

The model fitted closely to estimated teeth proportions at all leaf ranks and all tooth ranks (Figure 4b, and Supplementary Figures S9 and S10). In some cases, there was some discrepancy between model and data for later teeth, probably due to some teeth missed in larger leaves, as they tended to smooth out (Biot et al., 2016). Tooth initiation times estimated using the model showed a coherent trend across leaves, with a close-to-perfectly linear increase of initiation time as a function of leaf rank (Figure 4c). This pattern was consistent with the faster initial development of low-ranked leaves. In addition, there was also a trend for the time interval between successive teeth to increase. In *kluh* and *clfsep* L11, teeth developed sooner than those of Col-0, in accordance with the earlier growth observed in these mutants. However, the interval between successive teeth was conserved, suggesting independent regulation between the initiation of the leaf developmental program and its temporal schedule once initiated.

### 3.6. Morphogenesis of successive teeth

Using measurements of tooth width and height (Figure 4d), we examined how successive teeth grow over the contour of a given leaf (Figure 4e). When blade length was used as a proxy of time, distinct teeth seemed to grow at different rates. For instance, the first three teeth in Col-0 L11 seemed to grow in width following distinct dynamics (Figure 4f). To properly analyse tooth growth kinetics, the age of each individual tooth was determined as the age of the leaf minus the tooth initiation time. Note that this allows temporal registration of tooth developments, whether teeth are developing on the same or on different leaves. This procedure radically changed the perception of relative tooth development and revealed that the three teeth in Col-0 L11 actually followed similar

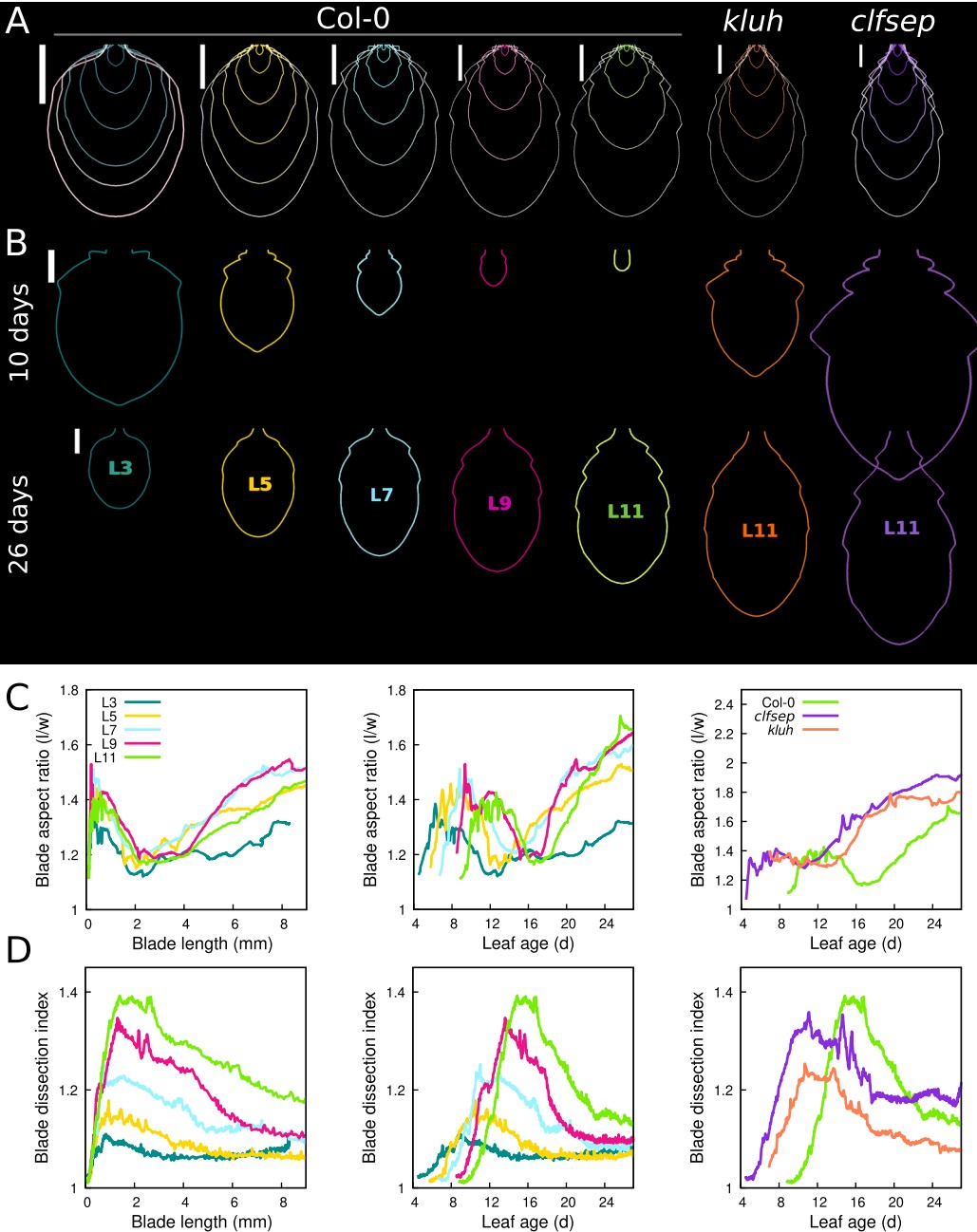

**Fig. 3.** Global leaf shape trajectories. (a) Selected contours from growth trajectories of wild-type and mutant leaves, at leaf ages of 220, 290, 360, 430, 500 and 570 hr. From left to right: L3, L5, L7, L9 and L11 in Col-0, L11 in *kluh*, and in *clfsep*. Scale bars: 2,500 μm. (b) Blade contours at 10 and 26 days after leaf initiation (top and bottom, respectively). Scar bars: 250 and 2,500 μm, respectively. (c) Blade aspect-ratio (length/width), for Col-0 leaves, as a function of either blade length (*Left*) or leaf age (*Middle*), and for wild-type and mutant L11 as a function of leaf age (*Right*). (d) Leaf blade dissection index (blade convex hull area/blade area), displayed as in panel (c).

dynamics for growth in width (Figure 4g). Similarly, dynamics of tooth height were initially superimposed. However, starting from 200 to 300 hr, the rate of growth in height started to decrease differently depending on tooth rank (Figure 4h). Computing aspect-ratio confirmed that, as previously shown, successive teeth are more and more pointed as the rank of apparition increases (Biot et al., 2016). However, plotting its dynamics as a function of tooth age further revealed a synchronisation across teeth, with a maximal sharpness reached 100 hr after initiation (see Figure 4i for L11 and Supplementary Figure S12 for other Col-0 leaves). These results suggested successive teeth in a leaf follow a common morphogenetic

program with graded differences from one tooth to the next and further demonstrated the considerable gain at expressing measures in proper chronological times.

We next compared the evolution of teeth of the same rank but developing at the margin of different leaves. Before 200 hr, Tooth 1 in Col-0 leaves all grew in width at comparable rates regardless of the leaf rank (Figure 4j). Beyond 200 hr, tooth growth in width slowed down all the more as leaf rank was low. To remove a potential effect of growth differences across leaves (Figure 2e), we normalised tooth width by blade length. Obtained measures displayed remarkably similar dynamics (Figure 4k). This further

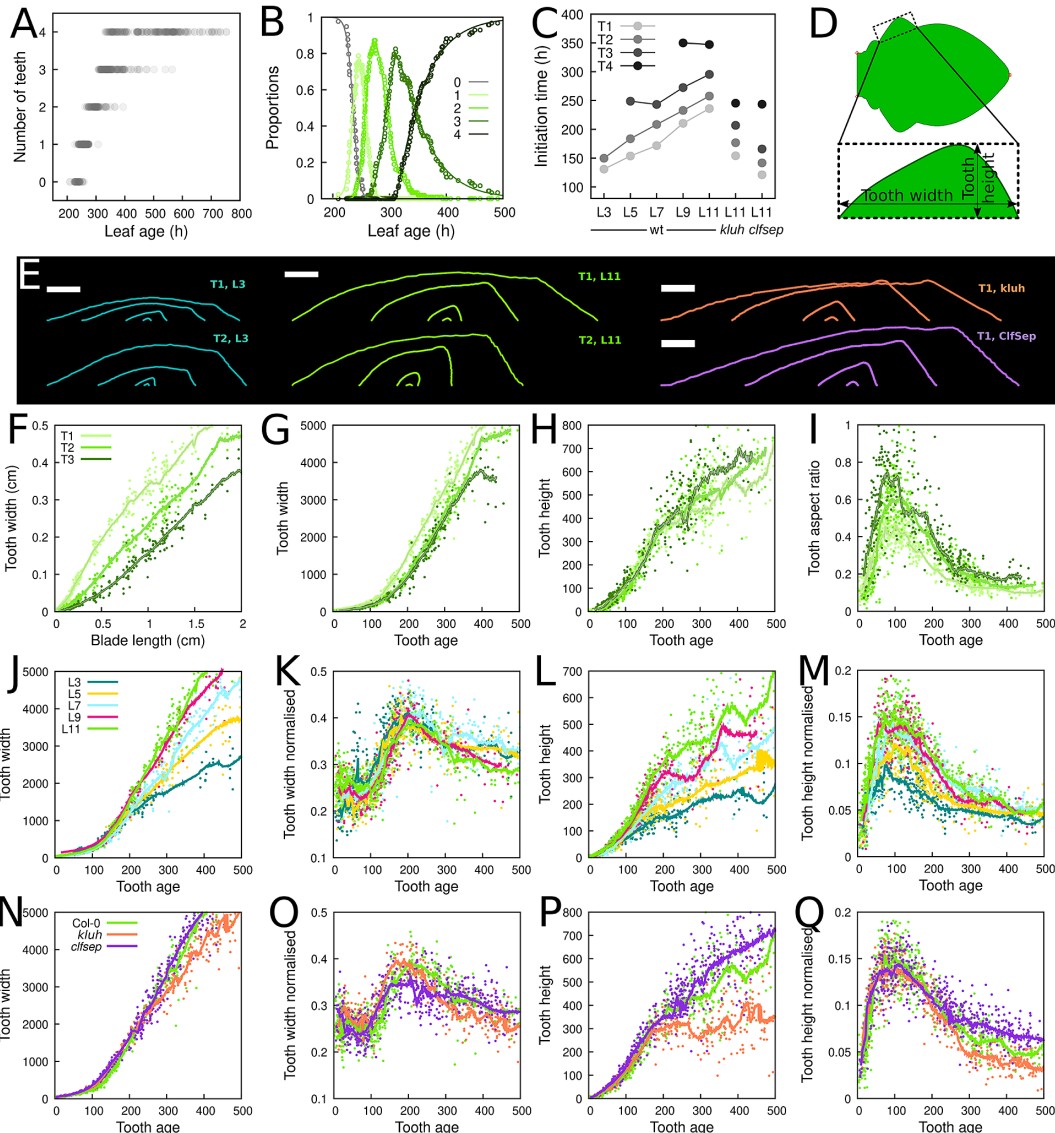

**Fig. 4.** Morphological analyses of leaf teeth. (a) Number of teeth in Col-0 on one side of L11, according to leaf age (opacity proportional to data density; data from both leaf sides were pooled). (b) Temporal dynamics of empirical (*Dots*) and model-predicted (*Lines*) proportions of teeth with different ranks. (c) Estimated teeth initiation times on one side of the leaves. (d) Measures of tooth shape. (e) Teeth contours in Col-0 L3 and L11 (top and bottom: Teeth 1 and 2, respectively), and in *clfsep* and *kluh* L11 (Tooth 1). Common selected tooth ages: 30, 110, 190 and 270 hr. Scale bar: 250 µm. (f–i) Measures of Teeth 1, 2 and 3 in Col-0 L11. Tooth width as a function of blade length (f) and of tooth age (g), tooth height (h) and aspect-ratio (height/width) (i) as functions of tooth age. (j-m) Tooth 1 measures in Col-0 leaves. Tooth width (j), tooth width normalised by blade length (k), tooth height (l) and tooth height normalised by blade width (m), as functions of tooth age. (n–q) Same quantification as in panels (j–m), but for L11 in Col-0, *clfsep* and *kluh*.

emphasised three distinct phases. Between initiation and 100 hr, the tooth grew in width at the same rate than blade in general, then faster until 200 hr, at which point it started to grow slower. Tooth 1 height dynamics showed early divergence between leaves, with a gradually increasing growth rate according to leaf rank (Figure 4l). After normalisation by blade width, we observed synchronised dynamics, with a relative height that increased with leaf rank, and in all cases the maximum was reached at ~100 hr (Figure 4m), when Tooth 1 was also the most pointy (Figure 4i and Supplementary Figure S12). These results show that tooth width and height do not evolve proportionally to blade size but instead follow independent dynamics, and suggest teeth on successive leaves develop according to a shared, graded pattern.

In *kluh* and *clfsep* mutant L11, early growth in width of Tooth 1 displayed the same dynamics as in Col-0. Then, starting from 200

hr, tooth lateral growth rate in *kluh* slightly decreased (Figure 4n), while in *clfsep* it remained similar to Col-0. This difference was likely a side-effect of leaf growth differences because it was mainly abolished after normalisation by blade length (Figure 4o). On the opposite, striking contrasts appeared when comparing tooth heights (Figure 4p). At 200–250 hr of tooth development, L11 Tooth 1 stopped growing in height in the *kluh* mutant, while it started to grow slightly faster in *clfsep* compared to Col-0. These effects were also apparent after normalisation of tooth height by blade width, which clearly showed the divergence in dynamics between the three genotypes from ~250 hr (Figure 4q). Altogether, these results highlighted how our dating methodology can finely dissect the spatiotemporal effects of mutations by providing precise quantification of affected morphological traits and temporal windows.

### 3.7. MorphoLeaf new release

The new version of MorphoLeaf software offers the possibility to enter estimated growth kinetics parameters [$L_\infty$, $t_{50}$ and $n$ in equation (2)]. An age is then automatically attributed to all individual organs, computed from blade length measures. This information is then stored in the measurement files generated by the application, thus allowing shape analyses as functions of time. Using these kinetics parameters, MorphoLeaf now also generates leaf growth trajectories based on standard chronological time instead of blade length.

## 4. Discussion

Based on a previous pipeline exploiting static image data to reconstruct organ shape dynamics during development (Biot et al., 2016), we show here how biological time can be recovered to generate accurate morphogenetic trajectories. The average, explicit shape of the growing organ is given at any timepoint, over the whole developmental period, with no limitation of duration (Supplementary Figure S1). The potential of our methodology was illustrated with the analysis of leaf morphogenesis in *Arabidopsis thaliana*, a challenging model because this organ undergoes both significant shape changes immediately after its initiation, and important size evolution, from microscopic to macroscopic scales. We highlighted notable, and sometimes subtle, events occurring at precise moments during growth, which have long-term effects on the organ shape. Our results stress the necessity to consider the whole developmental process to identify major determinants controlling the shaping of an organ.

### 4.1. A new method for reconstructing temporal dynamics

The problem of expressing time using a developmentally meaningful index is not new. In the case of leaf morphogenesis in plants, the most popular method is the PI that was developed to compare individual dynamics affected by temporal variability and to identify critical developmental timepoints (Erickson and Michelini, 1957; Meicenheimer, 2014). The PI is computed based on the rank and lengths of the first pair of leaves that are above and below some reference length, and the LPI is obtained by subtracting leaf rank from the PI (Erickson and Michelini, 1957). The major strength of the PI/LPI is to be linearly related to plant or leaf age and to only require simple length measurements on successive leaves until the reference length. However, the PI relies on the assumptions of exponential growth, constant relative growth rate and constant plastochron for successive organs, thus restraining its scope of application, even though variants have been proposed to relax some of the underlying assumptions (Chen et al., 2009; Hill and Lord, 1990). Our results stress that these assumptions may be violated, for example, when comparing mutants to wild-type leaves or when analysing successive teeth that do not appear at constant time intervals. By contrast, our approach only assumes, for estimating initiation times, that organs are initiated sequentially and that this order can still be inferred at later stages. It also relies on modelling growth curves using an invertible mathematical growth function to relate organ size to plant age. Many growth functions have been proposed to model plant growth (Yin et al., 2003; Zeide, 1993), and though we used Hill functions here because they fit to our data, alternative functions could be similarly used in other experimental situations. Our approach requires initial measurements for estimating initiation times and for calibrating growth curves.

As the PI/LPI method requires the collection of longitudinal data to check the validity of underlying assumptions (Meicenheimer, 2014), the associated experimental cost is comparable between the two approaches. Overall, we believe that the combination of model-based estimation of initiation times combined with the fitting of growth curves provides a widely applicable approach for temporal registration, comparison and analysis of developmental dynamics of plant organs.

### 4.2. Variations on a common morphogenetic program behind Arabidopsis leaf heteroblasty

Our analyses unmasked a repeated developmental scheme across leaves successively formed on plants of increasing age that display clear heteroblasty varying their sizes, overall shapes and serration levels at the mature stage. This developmental scheme showed both invariant aspects and graded variations acting at multiple levels.

For instance, at the organ level the relative growth rate at maximal growth and the triphasic elongation–rounding–elongation pattern of the shape dynamics are absolute invariants of the developmental scheme for all wild-type leaves, regardless their rank. In addition to these absolute invariants, several growth features exhibited similar dynamics up to a relative scaling or gradation with constant increment between consecutive leaves. In Col-0, growth kinetics functions of leaves showed incremental evolution according to leaf rank. Though the origin of these patterns remain unknown, one can speculate that the grading observed in final organ size may be due to a differential balance between cell division and expansion (Fleming, 2018; Vercruysse et al., 2020). Higher size of high-ranked leaves could result from a longer period spent at producing cells before entering the expansion/differentiation phase, thus exhibiting delayed growth acceleration, higher maximal growth rate, and larger final organs as shown by growth kinetics functions. Characterising the spatiotemporal patterns of transitions between cell proliferation and expansion in leaves of different ranks using cell cycle markers (Desvoyes et al., 2020) or regulatory genes (Nath et al., 2003) will help to evaluate this hypothesis.

At the local scale of margin teeth, we similarly revealed invariants and graded features. Tooth growth in width on different leaves showed a systematic biphasic pattern, with a common initial exponential growth up to 200 hr. Temporal synchronisation was also observed in the evolution of tooth shape, with a maximum sharpness systematically reached at about 100 hr. As in the case of leaves, other features exhibited graded patterns evolving incrementally with tooth or leaf rank. Tooth growth in width beyond the exponential phase followed a linear pattern with a slope increasing with leaf rank, and tooth height also exhibited an increasing rate with leaf rank.

We interpret these invariant and graded patterns as the readout of a common underlying developmental program. This module would be repeatedly invoked upon the apparition of successive organs or motives. Our mutant analysis shed some light on the modulations of this morphogenetic program. L11 of the *kluh* mutant initiates earlier than expected from wild type and has an overall growth kinetics resembling the one of wild-type leaves of rank lower than 11. This suggests that the *kluh* mutation has an heterochronic effect on leaf development, and therefore that the KLUH gene may impinge on the stage-modulation of the common morphogenetic program. On the contrary, because the growth kinetics of *clfsep* mutant leaves appears to be profoundly modified, the activity of actors of the common morphogenetic program may be affected in this background. While our study tested the robust-

ness of the leaf morphogenetic program in response to genetic perturbations or developmental transitions, it would be meaningful testing its behaviour in response to environmental perturbations that lead to leaf heterophylly (Li et al., 2019).

### 4.3. A surprising invariant growth pattern is associated with tooth formation

We observed that tooth growth in width was invariant during the 200 first hours between successive teeth in a given leaf, between leaves of different ranks and in the two mutant backgrounds we analysed. Such growth invariant contrasted with growth in height which was clearly variable in leaves of different ranks. As early growth in tooth width can be considered as a local measure of leaf margin growth at the site of tooth formation, this suggests all teeth develop in a comparable local context across leaves of different sizes and growth dynamics. At this stage, we cannot distinguish between cause and effect. Tooth initiation could for instance require a permissive local context characterised by a particular growth pattern, while on the contrary tooth initiation may be a conserved process that leads to a conserved growth pattern in its neighbourhood. Distinguishing between the two hypotheses would require precise growth characterisation at small scale, before and after tooth formation. Nevertheless, the observation of such an invariant pattern at the site of tooth formation exemplifies how tooth initiation is somehow connected to the global growth patterns.

Our observations point to additional invariants during tooth formation. The maximal pointiness of teeth reached at 100 hr was consistent with the beginning of a faster relative growth in width of the teeth compared with blade length. Similarly, the transition at 200 hr between an exponential and a slower, linear growth in tooth width was also in agreement with the transition at 200 hr to a lower relatively growth in width. It was remarkable that 200 hr was also the time at which tooth growth in both width and height in *kluh* mutant leaves started to diverge from the dynamics of the wild type. These temporal 'rendezvous' point towards the existence of critical points during development where some yet unknown mechanisms are specifically starting to operate. Investigating which particular biological processes are involved at these particular moments, for example, by quantifying the expression of selected genes of interest, could provide clues about the biological factors perturbed by the *kluh* mutation.

### 4.4. Uncoupling between global and local growth patterns

Ours results showed that successive teeth at leaf margin share synchronised growth dynamics even though organs grow at drastically different rates. At 100 hr of Tooth 1 development in Col-0, for example, L3 and L11 were 231 and 336 hr old, respectively, thus a difference of more than 100 hr. At the same Tooth 1 age, L11 in *clfsep* was already 31% longer than in Col-0 (1,330 vs. 1,013 μm long) and overall leaf shapes also differed. These observations thus highlighted a decoupling between the mechanisms responsible for global (whole leaf) and local (teeth) tissue growth during morphogenesis (Supplementary Figure S13).

### 5. Conclusion

Modelling (Kierzkowski et al., 2019; Runions et al., 2017), comparative genetic (Challa et al., 2021; Hay and Tsiantis, 2006) and transcriptomics (Du et al., 2018; Ichihashi et al., 2014) studies have shown that modulations within conserved regulatory modules and shared mechanisms can generate diversity in leaf shape. Similarly, our work suggests that conserved regulatory modules are operating in different leaves and teeth. The strength of our strategy is to provide quantitative support for this interpretation. Because precise morphogenesis quantification is a powerful means to identify, in time and space, specific processes that are impacted by mutations or perturbed growth conditions, our strategy based on the use of massive static data is a precious tool for the development of quantitative models and for the understanding of biological mechanisms operating in organ morphogenesis.

## Acknowledgements

This work has benefited from the support of IJPB's Plant Observatory technological platforms.

**Financial support.** This work has benefited from a DIM 2015 Région Paris Ile-de-France PhD scholarship for M.O. The IJPB benefits from the support of Saclay Plant Sciences-SPS (ANR-17-EUR-0007).

**Conflict of interest.** The authors declare no conflicts of interest.

**Authorship contributions.** M.O., E.B., P.A., P.L. and J.B. conceived and designed the study. M.O., N.A., A.M.-C.and P.L. conducted data gathering. M.O., E.B., P.A. and J.B. developed and implemented models and algorithms. M.O., P.A. and J.B. wrote the article with inputs from all authors.

**Data availability statement.** New version of MorphoLeaf application is available at http://morpholeaf.versailles.inra.fr. COPASI and R scripts used to estimate temporal calibration parameters are available at https://doi.org/10.15454/DPFU1T. Leaf datasets are available at https://doi.org/10.15454/BMELNY.

**Supplementary material.** To view the supplementary material for this article, please visit https://doi.org/10.1017/qpb.2022.23.

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
