## [Reviewer Report]

Dear Editor-in-Chief,

Following your invitation to Philippe Andrey for a submisison to Quantitative Plant Biology, we are pleased to submit the enclosed manuscript by Oughou et al., entitled “Model-based reconstruction of organ growth dynamics reveals invariant patterns in leaf morphogenesis”.

In this work, we propose a global strategy that we believe represent an important methodological breakthrough for quantitative analysis of organ morphogenesis, especially over long time periods. Long-term growth monitoring with time-lapse acquisition techniques indeed rises severe technological issues, among which phototoxicity, heating, manipulations, that can significantly impact on development and on specimen survival.

We previously proposed a strategy to reconstruct developmental dynamics from static images acquired on distinct individuals at different time-points (Biot et al., Development, 143, 3417-3428, 2016; to this day 32 citations in WoS). This approach was implemented in the freely distributed MorphoLeaf software, which has been successfully used in diverse topics such as morphogenesis (Gonçalves et al., Journal of Experimental Botany, 2017; Maugarny-Calès et al., PLOS Genetics, 2019) paleoclimate (Tanrattana et al., International Journal of Plant Sciences, 2020), eco-physiology (Adamo et al., Plants, 2020), morphometrics (Oso and Jayeola, Applications in Plant Sciences, 2021), or pedagogy (Guijarro-Real et al., in proceedings of INTED2019, 2019). However, this approach was limited because actual developmental time was lost, organ size being used as a temporal proxy.

Here, we introduce a model-based strategy to recover the temporal information that is lost with static data. We applied this new approach to reconstruct and quantify Arabidopsis leaf morphogenesis as a function of time from initiation to maturity. We reveal several invariants and common developmental patterns in the building of global leaf shapes and of local serrations. Our analysis of mutant leaves with contrasted shapes further shows specific impacts of mutations on these patterns, thus paving the way to future dissection studies of the underlying molecular determinants.

For all these reasons, we think our contribution could be of interest for a large audience and hope you will consider it is suitable for publication in Quantitative Plant Biology.

Looking forward to reading from you,

Yours sincerely,

Jasmine Burguet.

---

## [Reviewer Report]

*Comments to Author*: The authors present a model-based approach to reconstruct the spatiotemporal dynamics of leaf morphogenesis using static images. A Hill model was used to fit data between empirical observations, resulting in a reconstituted inferred construction of emergent leaf shape dynamics.

A series of data input into the model made use of Arabidopsis leaves both from wild-type and mutant backgrounds. The models did a good job at matching observed data points in terms of organ initiation and morphology.

The work is well described and appears to perform its stated purpose.

Questions about the work relate to drawing conclusions using inferred of data. Mathematical models are invoked to generate dynamics in the absence of their measurement. While this appears to be accurate based on the data presented, it remains unclear to what extent these inferred data can be treated as bona fide experimental observations. Can the authors provide evidence that data gaps filled by models are scientifically robust? Perhaps through the use of naïve/novel datasets not included in the model parameterization? Or using genuine time lapse datasets which have been performed on the same sample over time?

---

## [Reviewer Report]

*Comments to Author*:

I apologise sincerely for the time required for this review. We struggled to find reviewers and unfortunately not all reviewers we had secured actually delivered their review to date. I am thus proceeding with only one review to not hold up the process even further.

Reviewer 1 makes some salient points that need to be considered and addressed. In addition, I attach my own comments below.

The authors use empirical functions to interpolate between timepoints to describe growth. They build a basic model to capture leaf initiation as a function of time and use simple metrics to describe leaf shape. They use this approach to build reference growth stages with time and use this to assign an age to leaf data. Their approach works wells and fits time, developmental time, growth and morphology. Overall, the work is clearly described and presented.

The authors stress their use of static data. As a movie is just a sequence of static images and static images or static data are of course used all the time for dynamic models, it would help the reader to appreciate the work if the authors explained more clearly what their advance is here.

Was the model validated? For instance, could experimental data with known times and organ numbers be subsampled and used to reconstruct data that wasn’t used? Or could data be simulated from a model with different parameters and then the parameters inferred from the procedure?

It is not clear why the authors have chosen the functions they used. Could their approach be coupled to a more mechanistic approach?

The manuscript would benefit from some editing:

Line 22 ‘coordinate’ -> ‘coordinated’

Line 22 I wonder what ‘stereotyped’ adds here

Line 23 to avoid confusion with evolutionary process, perhaps change ‘evolution’ for ‘changes’

Line 25 ‘which’ -> ‘whose’

Line 39/40 ‘giving potentially easier access to variability’ -> ‘providing data from which the variability between samples can be assessed’.

Line 40 ‘More,’ -> ‘Another advantage is that’

Line 74 ‘precise’ -> ‘make precise’

Line 124-125: I was surprised by the use of a Hill function for modelling growth. Perhaps this could be motivated and/or contrasted with other growth functions (Gompertz or logistic growth model - see Rickett et al. (2015) BMC Systems Biology and references within for a discussion of such functions in the context of population growth models, which have the same functional form as organ and plant growth models). What is the mechanistic basis for the Hill function in the context of growth? What does a Hill parameter mean in this context? When the Hill function was fitted (eg Supplemental Figures 2 and 4), what were values of the inferred parameters and their estimated uncertainties?

Line 129-130: What is A? f(tmax) doesn’t seem to be a function of tmax – what is f’ and f? Please define and explain the notation better.

Line 131 Are L(a) and l the same? Perhaps avoid duplication of notation.

Line 141-142: Some explanation about how the user might choose L, n and t50 would be useful. How sensitive are the results to this choice?

Line 148: ‘under’ -> ‘within’

Line 149: ‘under’-> ‘in’

Line 149: Some motivation for using the stiff ODE solver RADAU5 would be useful.

Line 168-169: The symbol for the union of sets seems out of place here where the authors means ‘and’ of two conditions, not sets. This notation is potentially confusing.

Line 177-179: This is simple model that progress one stage to the next – some references to the many examples (eg chain length elongation) of other models that use the same equations could be considered.

Line 182: What is the parameter sk? What is the mechanistic basis of this function here? Please explain all parameters.

Line 190: Please define ranks.

Line 192: What is ‘empirical evolution’?

Line 218 What is ‘remarkable’ here needs further explanation.

Line 320: Was blade length used directly as a proxy or was the functional relationship taken into account? If not, the observations are hardly surprising.

Line 366-368: Dynamical models using static data are commonplace. It would help the reader to bring out exactly what the advance is here.

Line 494: There are perhaps similar statistical patterns of simple growth metrics but I didn’t find evidence in this statistical analysis that relates to ‘conserved regulatory modules’. This sentence should be removed or backed up.

---

## [Reviewer Report]

May 25, 2022

Dear Editor,

We thank you and the Referee for your appreciable proofreading of our manuscript “Model-based reconstruction of organ growth dynamics reveals invariant patterns in leaf morphogenesis”.

We took all your remarks and suggestions into account and, below, answered them carefully, point by point. In particular, we added three Supplementary Figures, to clarify the potential of the approach we propose (Figure S1), to motivate the choice of our calibration model (Figure S5), and for validation (Figure S11).

We hope you will now consider our manuscript in its present form suitable for publication in Quantitative Plant Biology.

Best regards

Jasmine Burguet.

---

## [Reviewer Report]

*Comments to Author*: In the manuscript entitled “Model-based reconstruction of organ growth dynamics reveals invariant patterns in leaf morphogenesis” Oughou et al. use computational analysis and modeling to derive leaf growth rates from static images of different leaves taken at different time points. As the authors describe, this is a powerful strategy to gain understanding of organ scale morphogenesis throughout the growth of the organ without the need for time consuming and possibly damaging live imaging. The authors have made an update to the MorphoLeaf software to incorporate these analyses. They use their pipeline to analyze the growth of wild-type Arabidopsis Col-0 leaves 1, 3, 5, 7, 9, and 11. They also analyze leaf 11 from the kluh mutant which has smaller leaves and clf sep3 double mutant which has larger leaves. Some of the interesting findings are that wild type leaves initiate at a 1-day interval. As have previously been seen, they verify kluh has a faster plastochron. The maximal growth rate of wild type leaves increases with subsequent leaves, such that leaf 11 has a higher maximal growth rate than leaf 3. Finally, they found that tooth 1 grew at comparable rates, regardless of which leaf it formed on, i.e. leaf 3 or leaf 11. It appears this is a useful analysis pipeline for inferring growth from static images.

Comments:

Abstract: The abstract should be more specific about the exact findings. The abstract says: “Using our strategy, we revealed invariant, iterated developmental schemes and critical time points in Arabidopsis thaliana leaf morphogenesis, suggesting conserved morphogenetic modules. We provide evidences that modules determining local features may act independently of global leaf growth. In addition, graded differences in growth dynamics and final shapes of successive leaves suggest continuous variations in module expression.” These statements are vague and should be clarified/replaced with the specific results and findings.

Introduction: The authors introduce the difficulties in live imaging, which is the current standard for growth analysis. The live imaging that they are discussing is generally done at cellular resolution so the cellular growth mechanisms giving rise to organ shape and size can be determined. Of course, this type of imaging is difficult and can only be caried out over a few days. However, the static imaging that the authors use for this manuscript only gives them organ contours, not cellular resolution. This level of imaging is more similar to the imaging performed with phenomics where images of the plants including leaves are captured throughout the life of the plant and leaf growth can be tracked. In this manuscript authors are dissecting the leaves, so they get a better contour and more clear morphology, which is an advantage. Adding some introduction or discussion of this pipeline relative to phenomics would be useful.

Materials and methods: Please add details of how the images were generated. Currently, it just says observed on a binocular microscope. The brand and model of the binocular microscope must be specified, and the camera, and any specific imaging parameters, such as magnification.

In Figure 2 G and H, it should be made clear on the Y-axis that this is the growth rate of leaf length so that someone skimming the figure quickly is not confused. Something like leaf length growth rate or growth rate in leaf length would solve the problem. Otherwise, the standard assumption is growth of leaf area.

Line 266-267: “For instance, the two final teeth on both sides of L3 were well-formed at day 10 while at the same age, the first tooth of L11 (a leaf that ultimately carries at least four teeth on each side, two more that L3) was barely present (Figure 3B).” The meaning of day 10 in this sentence and Figure 3B need to be clarified. Reading this sentence, I assumed day 10 meant days after the initiation of that leaf, i.e. for L11 10 days after L11 initiated. This measure has been carefully defined in the paper. This would be a fair comparison because both leaves had the same amount of time to develop. However, when I look at Figure 3B, it looks like the 10 days is in plant age and the L11 leaf is barely a primordium. Here it would be good to add a row showing 10 days after leaf initiation, so the morphology of leaves at the same stage can be compared.

---

## [Reviewer Report]

*Comments to Author*: This is a thorough and detailed revision. There are a few remaining minor points that have been raised that I would like to give you the opportunity to address.

---

## [Reviewer Report]

September 30, 2022

Dear Editor,

We thank you and the Referee for your appreciable second proofreading of our manuscript “Model-based reconstruction of whole organ growth dynamics reveals invariant patterns in leaf morphogenesis”.

We took all your remarks and suggestions into account and answered them carefully (see attachment).

In particular, the title changed according to your suggestion, and we significantly rewrote the Abstract, specifying more clearly our results.

We hope you will now consider our manuscript in its present form suitable for publication in Quantitative Plant Biology.

Best regards

Jasmine Burguet.

---

## [Reviewer Report]

*Comments to Author*: Dear Authors,

Thank you for taking all the suggestions on-board and updating your manuscript accordingly. I am very happy to accept your revision. Congratulations. Thank you for choosing QPB for your research. We hope to see further contributions from you in the future.

With thanks and best wishes

Richard